# Development of a Novel Bronchodilator Vaping Drug Delivery System Based on Thermal Degradation Properties

**DOI:** 10.3390/ph16121730

**Published:** 2023-12-15

**Authors:** Mariam Chaoui, Emmanuelle Fischer, Sophie Perinel-Ragey, Nathalie Prévôt, Lara Leclerc, Jérémie Pourchez

**Affiliations:** 1Mines Saint-Etienne, Université Jean Monnet Saint-Etienne, INSERM, Sainbiose U1059, Centre CIS, F-42023 Saint-Etienne, France; mariam.chaoui@emse.fr (M.C.); emmanuelle.fischer@etu.emse.fr (E.F.); sophie.perinel.ragey@univ-st-etienne.fr (S.P.-R.); nathalie.prevot@univ-st-etienne.fr (N.P.); leclerc@emse.fr (L.L.); 2Medical-Surgical Intensive Care Unit, CHU Saint-Etienne, F-42055 Saint-Etienne, France; 3Nuclear Medicine Unit, CHU Saint-Etienne, F-42055 Saint-Etienne, France

**Keywords:** vaping drug delivery systems, thermal degradation, bronchodilators, drug deposition

## Abstract

This work aims to investigate bronchodilator delivery with the use of different vaping drug delivery systems (VDDS) by determining the dose equivalence delivered in relation to different references: a clinical jet nebulizer, a pMDI (pressurized metered dose inhaler) and a DPI (dry powder inhaler). Three different bronchodilators were used (terbutaline, salbutamol hemisulfate, ipratropium bromide). The e-liquids contained the active pharmaceutical ingredient (API) in powder form. Two different VDDS were tested (JUUL and a GS AIR 2 atomizer paired with a variable lithium-ion battery (i-stick TC 40 W), 1.5 ohm resistance, and 15 W power). Samples were collected using a glass twin impinger (GTI). High-performance liquid chromatography (HPLC) was used to quantify the drugs. A next-generation impactor (NGI) was used to measure the particle size distribution. Terbutaline emerged as the optimal API for bronchodilator delivery in both VDDS devices. It achieved the delivery of a respirable dose of 20.05 ± 4.2 µg/puff for GS AIR 2 and 2.98 ± 0.52 µg/puff for JUUL. With these delivered doses, it is possible to achieve a dose equivalence similar to that of a jet nebulizer and DPI, all while maintaining a reasonable duration, particularly with the GS AIR 2. This study is the first to provide evidence that vaping bronchodilators work only with appropriate formulation, vaping technology, and specific drugs, depending on their thermal degradation properties.

## 1. Introduction

Inhalation therapy is a type of medical treatment involving the delivery of medication directly to the lungs through inhalation. It is commonly used to treat respiratory conditions such as asthma, chronic obstructive pulmonary disease (COPD), and cystic fibrosis. The three most common and predominant platforms for administering therapeutic agents as aerosols through inhalation devices include nebulizers, pressurized metered-dose inhalers (pMDI), and dry powder inhalers (DPIs). Undoubtedly, in addition to the above-mentioned treatments, other types of inhalation therapies are available on the market, such as soft mist inhalers (SMIs). Nonetheless, not all APIs (active pharmaceutical ingredients) are compatible with all inhalation devices due to challenges related to formulation, particle size, stability, and bioavailability. While pulmonary drug delivery is effective at administering medications directly to the respiratory tract, it is crucial to acknowledge that there are certain limitations associated with the utilization of the actual inhalation devices. For example, for pMDIs, it has been reported that approximately 50 to 76% of patients make at least one mistake when using these types of devices. Furthermore, 4 to 94% of patients with DPIs do not use them correctly, and 25% of them have never received training on how to use these devices. A low inspiratory flow rate (i), poor hand–mouth coordination (ii), and the sudden cooling of the canister are the main causes of non-compliance with inhaled treatments [1,2,3]. On the other hand, most jet nebulizers have the ability to aerosolize a wide range of drug solutions, delivering substantial doses with minimal requirements of patients; no skills related to mouth–hand coordination are required. It is crucial to acknowledge, however, that nebulizer-based treatments may be perceived as time-consuming and notably inefficient, particularly due to the significant amount of drug waste, often exceeding 50%. On average, only 10% of the prescribed dose reaches the lungs, a phenomenon attributed to factors such as the dead volume within the nebulizer or the release of aerosol into the environment during expiration [4,5]. In view of the aforementioned considerations, it becomes necessary to develop other delivery technologies such as the ones utilizing diverse aerosolization mechanisms, including those regarded as evaporation–condensation technology.

Notably, in 2012, the FDA approved the first product known as ADASUVE™, employing the Staccato^®^ technology. This system operates through rapid vaporization induced by heating a surface coated with a thin film of the drug. It is breath-activated and features a valve regulating airflow to attain the desired particle size [6,7,8]. The Staccato system generates an aerosol with small particle dimensions (with a mass median aerodynamic diameter in the 2 µm range for Loxapine), which is well suited for deep lung deposition and systemic administration [9]. This inhaler is portable and available in both single- and multi-dose configurations, depending on the specific drug product used [6]. The Staccato ADASUVE™ is marketed by Ferrer Therapeutics, and its formulation contains loxapine, designed for the management of agitation in individuals with schizophrenia and bipolar I disorder. In addition to ADASUVE™, there are three additional pharmaceutical products in the pipeline that utilize the Staccato delivery device: alprazolam (for epileptic seizures), apomorphine (for Parkinson’s disease), and granisetron (for cyclic vomiting syndrome) [10]. Another illustration is the ARIA™ pulmonary delivery technology, which comprises a heated capillary tube designed to vaporize the liquid that it contains. Upon inhalation, the vaporized substance condenses due to a temperature decrease [11]. This particular device was developed by Chrysalis Therapeutics, a division of Philip Morris, USA. Finally, another product is being developed by Windtree Therapeutics, which is called Aerosurf^®^; it employs capillary technology for enabling an evaporation–condensation approach [10]. Aerosurf^®^ successfully completed phase 2b clinical trials aimed at developing a non-invasive therapy utilizing KL4 surfactants for the treatment of respiratory distress syndrome in premature infants [12]. Another drug delivery system relies on rapid vaporization and utilizes carriers with low vapor pressure [11]. This device consists of a heating filament, a surface coated with a low vapor pressure carrier, and a therapeutic agent. Unlike previous devices, direct heating of the drug is avoided, thus preventing drug degradation. Instead, a low vapor pressure carrier is heated and vaporized, subsequently releasing the therapeutic agent [11].

In this context, ENDS (electronic nicotine delivery systems), commonly referred to as electronic cigarettes, have garnered attention as potential vehicles for drug delivery [5]. These devices also rely on evaporation–condensation technology. In brief, ENDS are triggered via an automatic flow sensor or manual pushbutton, which activates the device’s heating element, subsequently vaporizing the solution contained within it. Various consumer devices with diverse designs are available. When inhaled, air is drawn in through the inlet ports, passes over the heating element, and proceeds toward the mouthpiece. This airflow induces a reduction in vapor temperature, leading to the condensation of vapor into liquid droplets that are then introduced into the respiratory tract [13]. All things considered, ENDS have been characterized as a potential substitute for conventional inhalers. They offer enhanced patient convenience and the ability to generate micro-scale carrier droplets with a consistent concentration of the active pharmaceutical ingredient, as highlighted our previous studies [14,15,16]. To eliminate any potential confusion with conventional electronic cigarettes, and given that all our formulations are nicotine- and flavor-free, we will refer to the device as a VDDS (vaping drug delivery system) for the rest of this paper.

In our recent study, we examined the use of terbutaline, a short-acting β2 agonist (bronchodilator) primarily prescribed for patients with asthma and COPD, in combination with VDDS. The study revealed that dose equivalence could be achieved with 52 puffs, lasting approximately 15 min, which was comparable to a 5-minute nebulization session [14]. This finding underscores the importance of investigating the compatibility of other molecules, such as salbutamol and ipratropium bromide, when used in combination with VDDS devices. Therefore, the primary objective of this paper is twofold: Firstly, to identify molecules that are suitable for drug vaping and comprehend the critical factors to consider when selecting optimal drug candidates. Secondly, we aim to establish dose equivalence with existing treatments, such as nebulizers, pMDI, and DPI.

## 2. Results

### 2.1. Analytical Method Validation

Regarding Salbutamol hemisulfate, the linearity of the calibration curve was successfully validated within a range from 0 to 1400 µg/mL. The results met the acceptance criteria outlined in the ICH guidelines, with a linear regression coefficient of determination (R2) exceeding 0.999. Furthermore, it was determined that there was no matrix effect when PDO (1,3-propanediol) was combined with salbutamol hemisulfate. The limits of detection (LOD) and quantification (LOQ) were determined to be 2.60 and 7.90 µg/mL, respectively.

For Ipratropium bromide, linearity was validated within a range from 75 to 175 µg/mL. Similar to salbutamol hemisulfate, there was no matrix effect observed when PDO was combined with ipratropium bromide. The LOD and LOQ for ipratropium bromide were found to be 5.70 and 17.28 µg/mL, respectively.

A comprehensive summary of the method of validation, including additional details, can be found in the Appendix A. However, the validation method for terbutaline is not included, as it has been the subject of our research efforts over the past few years [14,15,16].

### 2.2. Bronchodilator-Delivered Doses for the Pulmonary Delivery Technologies Used as References (pMDI, PDI, Jet Nebulizer)

The respirable doses of various drugs administered through different inhalation methods were assessed. Terbutaline delivered via the Cirrus 2^TM^ nebulizer (Intersurgical, Marne-La-Vallée, France) resulted in a respirable dose of 1037.62 μg out of a total of 5000 μg. In the case of terbutaline administered through a DPI, the estimated respirable dose was 921.13 μg out of 4000 μg, equivalent to 8 doses of 500 μg. Salbutamol nebulization achieved a respirable dose of 1834.19 μg from a 5000 μg dose. Similarly, salbutamol delivered via pMDI yielded a respirable dose of 160.52 μg out of 800 μg, corresponding to 8 doses of 100 μg. For ipratropium bromide, the Cirrus 2^TM^ nebulizer (Intersurgical, Marne-La-Vallée, France) achieved a respirable dose of 416.80 μg from a 1000 μg dose, while the pMDI delivered 238.76 μg out of 500 μg, equivalent to 20 doses of 25 μg.

### 2.3. Bronchodilators Delivered Dose: VDDS versus Nebulizers, PDI, and pMDI

Terbutaline

In light of the delivered doses obtained via conventional treatments, the results highlighted the dose equivalency using VDDS. The VDDS (GS AIR 2) consistently delivers terbutaline at an average of 20.05 ± 4.2 µg/puff, implying that approximately 50 puffs would be necessary to achieve a dose equivalent to nebulization. Conversely, for a DPI with a recommended daily dose of 8 puffs, 46 puffs with the GS AIR 2 would suffice for dose equivalence. In contrast, the JUUL VDDS provides terbutaline at an average of 2.98 ± 0.52 µg/puff, requiring approximately 347 puffs to reach a dose equivalent to that of a jet nebulizer. Similarly, for a DPI, 309 puffs with the JUUL are required for dose equivalence.

These findings underscore the significant variations in dose delivery efficiency between VDDS and conventional treatments.

Salbutamol

The results indicate that salbutamol can indeed be delivered using the GS AIR 2 and JUUL devices. However, it is important to note that the quantity of salbutamol delivered was significantly low, precluding the establishment of a dose equivalent to that derived from conventional treatments (Table 1).

Ipratropium bromide

Based on the findings, it was determined that ipratropium bromide was not successfully delivered through any of the tested VDDSs (Table 1).

### 2.4. Delivery Time: VDDS versus Nebulizers, PDI, and pMDI

In order to determine the time delivery, it is important to distinguish between the two durations mentioned in Figure 1 and Figure 2. Duration 1 exclusively considers the recommended puff duration of 3 s. On the other hand, duration 2 encompasses not only the puff duration but also the inter-puff interval of 30 s, in accordance with the AFNOR norm XPD-90-300-3.

It is worth mentioning that in real-life usage of “ENDS”, users typically do not adhere to a specific inter-puff interval.

Only terbutaline will be addressed in this section, as it is the sole API found to be compatible with the tested parameters.

When comparing the dose equivalences of the two VDDS (GS AIR 2, JUUL) to that of a jet nebulizer, we observed varying durations for achieving equivalent doses. For the GS AIR 2, it took approximately 2.50 min for duration 1 (considering the recommended puff duration of 3 s) and approximately 27 min for duration 2 (which includes the puff duration and a 30-second inter-puff interval per the AFNOR norm XPD-90-300-3). In contrast, the JUUL required approximately 17.35 min for duration 1 and approximately 3.17 h for duration 2 to achieve a dose equivalent to that of a jet nebulizer. When comparing the GS AIR 2 and a DPI device (each puff lasting around 10 s), dose equivalence is reached in approximately 2.3 min for duration 1 and approximately 24.80 min for duration 2. Similarly, the JUUL, when compared to a DPI, requires approximately 15.45 min for duration 1 and approximately 2.82 h for duration 2 to achieve dose equivalence. These findings underscore the significant differences in time required to attain dose equivalence between the examined devices and methods of delivery.

### 2.5. Particle Size Distribution: VDDS versus Nebulizers, PDI, and pMDI

The tested nebulizer (Cirrus 2^TM^) (Intersurgical, Marne-La-Vallée, France) produces particles with a mass median aerodynamic diameter (MMAD) quite similar to that of the GS AIR 2. Specifically, the MMAD values for the jet nebulizer and GS AIR 2 are 1.74 ± 0.07 µm and 1.41 ± 0.03 µm, respectively. Upon examining Figure 3, it is evident that the Cirrus 2^TM^ (Intersurgical, Marne-La-Vallée, France) exhibits an additional population of particles with seemingly larger sizes, a characteristic not observed with the GS AIR 2. This discrepancy is anticipated, considering that electronic cigarettes inherently possess the ability to generate smaller particles on the micron scale. This smaller particle size is advantageous, as it enhances the potential for optimal deposition and penetration into the respiratory tract, thereby increasing the likelihood of effective drug delivery.

The findings of this study demonstrate that VDDSs are capable of delivering drugs within the microgram range per puff, making them potentially suitable for medications with prescribed dosages within this range. These devices offer several advantages, including their ability to generate an optimal MMAD in the micron range (around 1.4 μm). Furthermore, users can easily adjust the emitted dose by manipulating the power output or flow intake, as previously established in our earlier studies [15,16]. The MMAD range achieved by this formulation and device (GS AIR 2) closely aligns with 1.5 μm, which, as suggested by Usmani et al., represents the optimal size for achieving the highest total lung deposition, accounting for approximately 56% of the total emitted dose [17].

**Table 1 pharmaceuticals-16-01730-t001:** Characterization of terbutaline, ipratropium bromide, and salbutamol hemisulfate (salbutamol Sulfate) administered using different inhalation devices (nebulizer, pMDI, and (GS AIR 2, JUUL as vaping drug delivery systems (VDDSs).

		Nebulizer	DPI or pMDI	GS AIR 2	JUUL
Terbutaline	Delivered dose	1037.62 ± 172.00 µg	921.13 ± 220.37 µg	(20.05 ± 4.2 µg/puff)	(2.98 ± 0.52 µg/puff)
	MMAD	1.74 ± 0.07 μm (GSD 1.70 ± 0.08)	4.5 ± 0.20 μm (GSD 1.6 ± 0.30) [18]	1.41 ± 0.03 µm (1.44 ± 0.06)	Too low to be determined
Ipratropium bromide	Delivered dose	416.80 ± 73.74 µg	238.76 ± 54.73 µg	-	-
	MMAD	1.70 ± 0.12 µm (GSD 1.53 ± 0.03)	0.9 ± 0.00 μm (GSD 1.8 ± 0.00) [19]	-	-
Salbutamol hemisulfate	Delivered dose	1834.19 ± 138.78 µg	160.52 ± 15.31 μg	Too low to be determined	
	MMAD	1.73 ± 0.03 µm (GSD 1.73 ± 0.04)	5.2 ± 1.30 μm (GSD 4.8 ± 1.60) [20]	-	-

## 3. Discussion

### 3.1. The Choice of Formulation

Terbutaline and salbutamol, though not identical in chemical composition, share structural similarities as members of the beta-2 adrenergic agonist class of pharmaceuticals. In the course of our investigation, we discerned noteworthy distinctions in the delivery of terbutaline when administered via various VDDS, revealing significant variations in the emitted doses. Conversely, salbutamol hemisulfate exhibited limited delivery efficacy through VDDS, rendering the establishment of dose equivalence challenging.

This observed discrepancy between terbutaline, characterized as a salt-free molecule, and salbutamol, associated with hemisulfate, underscores the pivotal role of salt forms in API delivery. In pharmaceutical formulations, salt forms are often preferred due to their augmented solubility, stability, and bioavailability, especially in terms of addressing issues of poor aqueous solubility [21]. Nevertheless, when considering the context of VDDS, the selection of a salt-free API emerges as a crucial determinant.

Notably, a study conducted by Buonocore et al. [22] corroborated these distinctions, revealing that salt-free APIs yield higher emitted doses for VDDS delivery compared to salbutamol sulfate. The solubility characteristics of salbutamol include its high solubility in ethanol, limited solubility in water, and very good solubility in chloroform [23]. These properties rationalize the utilization of ethanol in the formulation, as mentioned by Buonocore et al., further accentuating the importance of judicious API form selection when utilizing VDDS to optimize emitted doses. Within this framework, opting for a salt-free API becomes an indispensable consideration for achieving higher emitted doses through VDDS.

When utilizing conventional ENDS for aerosol generation, the e-liquid comprises one of propylene glycol (PG), vegetable glycerol (VG), or a combination of both with varying ratios.

These two humectants are on the FDA’s Generally Recognized as Safe (GRAS) list of food substances [24,25]. However, PG has been shown to have irritant effects on the respiratory tract [26]. In addition to this possible irritation, PG and VG may decompose, leading to the production of toxic aldehydes. The formation of these aldehydes, such as acrolein, formaldehyde, and acetaldehyde, is one of the biggest concerns associated with ENDS use to date, even if the quantities are very low compared to those generated via tobacco combustion [27,28].

It is worth noting that a higher concentration of PG in the PG/VG mixture is associated with an increased delivered dose of nicotine [29].

On the other hand, we have PDO, known as Vegetol^®^, which is another humectant used in e-liquids. PDO has a number of uses: cosmetics, food processing, polymers, etc. [30]. It is used in food as a substitute for PG. The FDA also designates it as GRAS. As an e-liquid, PDO offers a number of advantages over PG and VG: better taste properties, lower thermal degradation, and aerodynamic properties comparable to those of a PG/VG mix [31].

In the aforementioned study conducted by Buonocore et al. [22], it is notable that their formulation included both PG and VG. This formulation yielded significant results, achieving a delivery dose of 245 μg/puff for salbutamol, with a MMAD falling within the range of 1.2–1.4 μm.

In our approach, we opted to use PDO as the preferred solvent due to its safety profile for patients. This choice was made to avoid the use of potentially irritating substances in conjunction with drugs intended to treat respiratory diseases.

Considering all the points discussed, it becomes evident that the selection of the e-liquid used plays a crucial role in determining both the emitted dose and the aerodynamic diameter.

Hence, it is important to consider both the choice of API form and the composition of the e-liquid when formulating a product intended for use in conjunction with VDDS.

The visibility of fog or aerosol produced by ENDS containing PDO compared to those containing VG/PG mixtures can vary. Both PDO and VG/PG are commonly used as carrier liquids in ENDS to create the visible vapor that simulates smoke.

While VG/PG mixtures are more traditional and widely used in e-liquids, PDO is a newer alternative. PDO is known for its potential to produce a smoother and less irritating aerosol compared to traditional VG/PG mixtures. However, the visibility of the aerosol may still be an issue, as it creates a visual effect similar to that of VG/PG mixtures.

The extent of the visible fog depends on various factors, including the concentration of PDO in the e-liquid, the device’s design, and the vaping conditions. Ultimately, both PDO and VG/PG mixtures have the capacity to produce visible aerosol, but the specific characteristics and appearance may differ based on the composition and formulation of the e-liquid.

During our experiments, we observed variations in visible fog production, particularly noting a more pronounced fog when using the GS AIR 2 compared to the JUUL. This discrepancy underscores the influence of the type of device used, including its technical features, such as power, resistance, tank capacity, etc. The specific design and features of each device can significantly impact the aerosolization process, affecting the visibility and density of the produced fog. Therefore, when comparing VDDS to different carrier liquids, like PDO or VG/PG mixtures, it is crucial to consider not only the liquid composition but also the device’s characteristics, as they play a pivotal role in determining the visual output during vaping.

### 3.2. The Choice of Device

Based on the findings related to terbutaline, it is evident that a simple alteration in the VDDS used can lead to significant changes in the delivered dose, with a factor of 6.6 observed between GS AIR 2 (20.05 ± 4.2 µg/puff) and JUUL (2.98 ± 0.52 µg/puff). This variation can be attributed to the power settings, where GS AIR 2 operates at a fixed 15 W, while JUUL, as per the manufacturer’s specifications, operates between 6 and 8 W [32]. This power differential provides a clear explanation for the differing terbutaline doses delivered. In line with previous research [15] into terbutaline, it has been demonstrated that an increase in VDDS power levels results in a proportional increase in the delivered dose.

Moreover, the design of the atomizer exerts a pivotal influence in terms of altering the delivered dose. A study conducted by Chaoui et al. shows that, even when maintaining identical API concentrations and VDDS power levels, merely changing the atomizer design can cause the delivered dose to switch from 9.6 µg/puff to 16.72 µg/puff [14].

### 3.3. Thermal Stability

As per the results presented in Table 1, it is evident that successful drug vaping is achievable with only one API, namely terbutaline. Beyond considerations related to formulation, particularly with respect to salbutamol, it is imperative to take into account the issue of thermal degradation. The thermal degradation of the drug using VDDS mainly depends on two essential elements:The heating temperature of the VDDS;The boiling point of the drug used.

In our first case, we used terbutaline with a boiling point set at 419 °C [33], and the heating temperature of the VDDS was set at 315 °C. The boiling point of PDO was set at 210–212 °C [30]. From these parameters, we can understand that terbutaline does not have a vapor state; therefore, it is delivered.

On the other hand, for ipratropium bromide, the boiling point is set at 230 °C [34], which is very close to the boiling point of PDO. Both substances experience the vapour state. This is the reason why nothing is delivered with the use of VDDS.

However, for salbutamol hemisulfate, the boiling point is set at 382 °C [35]. Its value, in this case, is superior to the heating temperature of VDDS. We expected the molecule to work in the same way as terbutaline. However, the molecule was barely detectable, which means that the substance seems to be less stable and extremely sensitive to heat compared to terbutaline.

### 3.4. Some Limitations

Indeed, VDDS are compatible with diverse patient profiles encompassing varying anatomical and lung function characteristics. Nevertheless, due to the distinct nature of this device compared to conventional treatments, it is imperative that patients undergo comprehensive training to ensure its correct usage. Indeed, advising users about the potential risks associated with vaping technology is crucial, particularly with regard to the conditions leading to the “dry puff” that promotes aldehyde production [36]. In addition, it has been found that intense and frequent inhalation contributes to greater aldehyde formation, mainly due to the greater consumption of e-liquid. As highlighted in a study performed by T. R. Sosnowski et al., high doses of PV/VG can be detrimental to pulmonary surfactant [37].

It is important to note that some VDDS may have higher aerodynamic resistance than DPIs, which may be a negative point for patients with weak respiratory function. Indeed, patients with conditions such as COPD or asthma often experience reduced lung function. High aerodynamic resistance in VDDS could impede the inhalation process in these individuals, making it more challenging to effectively administer the intended dose. This limitation may result in insufficient drug delivery to the lungs, impacting the therapeutic efficacy of the inhaled medication [38]. Consequently, the importance of training persists, particularly for individuals unfamiliar with vaporized delivery methods. Another crucial factor to take into account is the utilization of low power. In the VDDS employed in our study, the power output did not exceed 15 W. Consequently, we anticipated the absence of detectable volatile organic compounds (VOCs), as the heightened risk of VOC emissions primarily arose when excessive power was applied to the atomizer coil, as noted by Stephens et al. [39]. To substantiate this observation, a study conducted by Gillman et al. systematically investigated various devices operating at different power levels, ranging from 5 to 25 W, with the aim of elucidating the dynamics of VOC generation. Their findings demonstrated that at a power setting of 15 W, VOC emissions were minimal. Ultimately, the study’s conclusion underscored the close association between VOC emissions, the power employed, and the specific device type employed [40].

As ENDS are strictly prohibited aboard airplanes, in work places, and within numerous public places, the utilization of VDDS as a medical device will be a challenge, since it will be hard to distinguish between the two devices. For this specific reason, it will be essential to raise public awareness of the distinct identity of VDDS as a vaping device exclusively intended for medical applications, such as the treatment of respiratory diseases.

It would be useful to develop a VDDS that can help to visually distinguish between the two types of vaping devices. Devices intended for therapeutic purposes could have, for example, clear and distinctive labeling indicating their medical status, including terms such as “Medical Device” or “For Therapeutic Use”. A specific color could also be reserved for devices designed for therapeutic use, enabling their immediate visual identification. A unique physical design, possibly inspired by traditional medical devices such as pMDIs or DPIs, could also be used to visually differentiate them.

## 4. Material and Methods

### 4.1. VDDS Tested and Aerosol Generation

We conducted tests on two distinct VDDS, namely JUUL^®^ and the GS AIR 2 atomizer. Both of these devices come with reusable kits, featuring either a cartridge (in the case of JUUL) or a tank system (GS AIR 2). Due to the fact that the cartridge provided with the JUUL’s system contains components that differ from our formulation, a necessary step was emptying/cleaning the cartridge from its existing product and subsequently refilling it with our own formulation.

The GS AIR 2 atomizer (Eleaf, Shenzen, China), battery (iStick TC40 W, Eleaf), charger, and a variable wattage resistance of 1.5 ohm were purchased from a retail store named Vapostore in Saint-Etienne, France, that is dedicated to selling vaping products. JUUL^®^ was bought from the official manufacturer’s website (https://www.juul.com (accessed on 12 October 2023)). The GS Air 2 atomizer has a Pyrex tank with a capacity of 2 mL. It is characterized by the following features: diameter—14 mm; length—70 mm; connection—type 510; compatible with resistors—GS Air 2, 1.2 ohm/GS Air 2, 1.5 ohm/GS Air 2, 0.75 ohm. It is also equipped with an adjustable lithium-ion battery (iStick TC40 W) with a great autonomy of 2600 mAh and a maximal power of 40 W. A fixed power of 15 W was used for the experiments. GS AIR 2 resistance (1.5 ohm) was composed of Kanthal, i.e., a heating wire, which enables the e-liquid to be heated up, thus producing vapor. Indeed, the latter is created by heating the cotton soaked in the e-liquid present in the VDDS. The shape of the resistance is cylindrical in the majority of cases.

On the other hand, JUUL’s system does not require changing the resistance, since it has an integrated system that offers the combination of resistance (1.6 ohm), mesh, and a reservoir all in one piece known as the cartridge. The power used for this device is between 6 and 8 W.

Before each experiment, the batteries were fully charged. In order to avoid any form of e-liquid overfilling, GS AIR 2 was filled up to 1.5 mL, and JUUL’s cartridge was filled up to 0.5 mL (capacity: 0.7 mL).

To generate an aerosol, a puffing machine (PE_MOD, Burghart, Germany) was used. It is an aerosol pump consisting of two identical pump modules connected via a common aerosol valve. Each module is a simple linear piston pump with a glass cylinder. The alternating operation of the two pump modules results in an almost continuous aerosol flow. A motor powers these pumps to mimic inspiration and expiration, as performed with a vaping device, thus ensuring the repeatability of the results (in terms of the delivered dose per puff, puff duration, etc.). Indeed, the aerosol pump is precise and controllable, being operated via a separate touch screen terminal. To fulfill the specific requirements of this study, the puffing regimen adhered to the guidelines outlined in the AFNOR Standard XPD-90–300–3 [41]. This norm dictated particular parameters that were strictly followed, including a puffing volume of 55 mL, a duration of 3 s for each puff, an inter-puff interval of 30 s, two series of 20 puffs, and an inter-series interval of 5 min.

It is imperative to note that the puffing machine is a device only designed for laboratory use. The intended use is to draw air, tobacco smoke, or another substance into the inlet of the valve and direct the aerosols to a fitting at the outlet of the valve so that the aerosol is released in the GTI or the NGI. This can be carried out in a series of puffs with long intervals or in continuous operation to create a near-continuous flow of aerosol.

### 4.2. Active Pharmaceutical Ingredients and e-Liquid Refill Formulation

The three main molecules prescribed for asthma and COPD are salbutamol sulphate, terbutaline sulphate, and ipratropium bromide. This is the reason why, in this study, we primarily focused on these three APIs. The first two molecules are both short-acting inhaled beta-2 mimetic bronchodilators, which ultimately cause the relaxation of the airway muscles, leading to bronchodilation and, thus, increased airflow. Ipratropium bromide is an inhaled anticholinergic bronchodilator, and it prevents bronchoconstriction and causes the airways to dilate. These molecules are administered via inhalation, and their effect is only temporary. They are not curative treatments with the aim of eliminating the cause but palliative treatments with the aim of relieving the symptoms of the disease. Thus, they are indispensable treatments in the daily lives of patients with this type of condition. They are mostly delivered via the pulmonary pathway.

In the context of our experiments conducted using a jet nebulizer (Cirrus 2^TM^) (Intersurgical, Marne-La-Vallée, France), pMDI, and DPI, we utilized the following commercially available inhaled medications for testing, which were all sourced from the French market:Bricanyl^®^ (Terbuhaler) 500 µg/dose: AstraZeneca (Dunkerque, France);Terbutaline^®^ (nebulizer solution) 5 mg/2 mL: Arrow, (Lyon, France);Salbutamol^®^ (nebulizer solution) 5 mg/2.5 mL: Teva (Courbevoie, France);Ventoline^®^ (pressurized metered-dose inhaler) 100 µg/dose: GSK (Evreux, France);Ipratropium^®^ (nebulizer solution) 0.5 mg/2 mL: Zentiva (Paris, France);Atrovent^®^ (pressurized metered-dose inhaler) 20 µg/dose: Boehringer Ingelheim (Paris, France).

Given the varying commercial availability of the APIs used in different inhaler devices (DPI and pMDI), we conducted tests using terbutaline sulphate via both a jet nebulizer and DPI. Subsequently, we examined salbutamol sulphate and ipratropium bromide using a pMDI and a nebulizer.

Nebulizers’ experiments: all nebulizations were realized using the Cirrus 2^TM^ nebulizer from Intersurgical (Marne-La-Vallée, France) combined with its compatible compressor.

To determine the nebulization time before formal experiments, the tested sample was placed in a dried nebulization cup connected to the compressor, and we continuously launched the nebulization until no more aerosol formed. The whole process was timed using a stopwatch and recorded (any fraction of one minute was counted as one minute to ensure complete nebulization). The volume and nebulization times of all samples are summarized in Table 2.

On the VDDS side, the powder forms of salbutamol hemisulfate, terbutaline, and ipratropium bromide were dissolved in saline solution (Miniversol 0.9% NaCl from Aguettant (Lyon, France)) to create stock solutions. These stock solutions had concentrations of 40 mg/mL, 16 mg/mL, and 2 mg/mL, respectively, for terbutaline, salbutamol hemisulfate, and ipratropium bromide.

The liquid refill was prepared by mixing the stock solution with PDO. All the solutions were prepared with the same ratio (87.5% PDO:12.5% stock solution of API) (Figure 4).

Throughout the VDDS experiments, the choice of the tested concentrations mainly relied on the actual concentrations used in the jet nebulizer (Cirrus 2^TM^): 2.5 mg/mL (Terbutaline), 250 µg/mL (ipratropium bromide), and 2 mg/mL (salbutamol hemisulfate). Each API was tested separately.

In order to ensure uniformity and homogeneity, the solutions were gently mixed. The preparation was performed at room temperature to simulate real-world usage, not exceeding a temperature of 25 °C.

All vials were kept in foil pouches and protected from light. Prior to each experiment, stock solutions and refill liquids were freshly prepared (ensuring that they were used on the same day).

It is important to emphasize that in this study, the inclusion of conventional treatments (jet nebulizer, pMDI, DPI) in conjunction with VDDS was specifically carried out to simplify the process of establishing a dose equivalent to that of the VDDS.

### 4.3. Validation of Chromatographic Conditions and Analytical Methods

The chromatographic framework comprised the Shimadzu HPLC system (Shimadzu, Tokyo, Japan) equipped with a pump (LC-40B XR), a UV-detector (SPD-40D), a degassing unit (DGU-403), an autosampler (SIL-40C), a column oven (CTO-40C), a controller (CBM-40lite), and a solvent reservoir. Data acquisitions were rendered using Lab solutions software Ver.5.6 (Shimadzu, Kyoto, Japan).

A Hypersil GOLD™ aQ C18 column (150 × 4.6 mm, 3 μm) fitted with a security guard column (10 mm, 3.0 mm ID, 5 µm particle) was used for the chromatographic separation of terbutaline and salbutamol hemisulfate. Indeed, these two APIs have similar mobile phase compositions but different chromatographic conditions.

For ipratropium bromide, a different HPLC column was utilized (SUPELCOSIL LC-18, 150 × 4.6 mm, 5 µm).

All the chromatographic conditions practiced for drug quantification are summarized in Appendix A.

The validation of an analytical method is a critical process, ensuring that it is fit for its intended purpose and capable of producing accurate and reliable results. For this specific reason, we used an analytical validation method based on ICH Q2. Five analytical performance characteristics were to be tested: linearity, matrix effect, repeatability, recovery, and the limit of quantification (LOQ)/detection (LOD).

The LOD and LOQ values were numerically decided based on the standard deviation of the response and the slope.

Linearity was measured using the least squares regression model to assess the slope, y-intercept, and correlation coefficient. The calibration curves were obtained from various terbutaline, salbutamol and ipratropium concentrations ranging from 0 to 2500 µg/mL, 0 to 1400 µg/mL, and 75 to 175 µg/mL, respectively. Each injection was performed six times.

### 4.4. Drug Fraction Separation

A glass twin impinger (GTI) (MC2, Lyon, France) is a device used to collect generated aerosols for analysis, and it is mainly used in the pharmaceutical industry to test the effectiveness of a specific drug by separating the respirable dose from the non-respirable dose.

The device consists of two chambers (upper and lower) connected in series. The upper chamber represents the non-respirable dose, and the lower one contains the respirable dose. Both chambers should contain a liquid medium, and it comes in different forms (water, buffer solution) depending on the tested chromatographical conditions. The upper chamber is filled with 7 mL of liquid, while the lower chamber contains 30 mL of liquid.

For our terbutaline and salbutamol hemisulfate experiments, deionized water was employed for both chambers. In the case of ipratropium bromide, the chambers were filled with the identical solvents utilized for the mobile phase (70% phosphate buffer:30% acetonitrile).

To ensure accurate particle collection, the vacuum pump connected to the GTI was first calibrated using a flowmeter (DFM3, Copley Scientific, Nottingham, UK); the required flowrate was set at 60 ± 5 L/min, as validated by the European Pharmacopeia (Ph. Eur.).

When the air containing the particles or aerosols passed through the GTI, the liquid in the upper chamber captured the particles. The air then passed through the lower chamber, which captured any remaining particles. To be more explicit, this model was based on a lung model and allowed the separation of two fractions according to a cut-off diameter of 6.4 μm. The lower part of the setup was considered to be the lower airways, while the upper part of the setup was considered to be with the upper airways, which did not allow the administration of the API at the site of action (lungs).

Once the sampling period concluded, to ensure consistent volume across all experiments, the solution collected in the upper impinger was transferred into a 10 mL flask and topped up to reach the desired total volume. Similarly, the protocol was applied to the lower impingement chamber, where the collected solution was placed into a 50 mL flask and adjusted to achieve the desired final volume. At the end, the liquid from each flask was subsequently quantified using HPLC (Figure 5).

It is important to note that the puffing machine generator was exclusively connected to the VDDS. Other inhaler devices were directly joined to the GTI without intermediaries.

### 4.5. Drug Deposition: Aerodynamic Diameter

The next-generation impactor (NGI) (Copley Scientific, Nottingham, UK) is an impactor used to evaluate the properties of the aerodynamic distribution of the generated particles, such as MMAD (mass median aerodynamic diameter) and GSD (geometric standard deviation). The separation of the particles depends on the velocity and aerodynamic particle size. The NGI consists of different stainless-steel trays with different-sized holes placed above a reservoir. The airflow passes through the device while allowing the particles to move and reach the trays, where they acquire the associated cut-off diameter.

This impactor was designed to meet the regulatory requirements for the testing of inhalers and is widely accepted by regulatory agencies, such as the United States Food and Drug Administration (FDA) and the European Medicines Agency (EMA). The NGI can be used in the testing of a variety of inhalable drug products, including pMDI, DPI, and nebulizers. It is a crucial tool in the development of effective and safe inhalable drug products.

The suitable flow rate primarily depended on the specific device being tested. For nebulizers, a recommended flow rate of 15 L/min was advised. Consequently, our experiments involving both VDDS and nebulizers were conducted at this specified flow rate.

Generally, it is important to note that puffs generated via the VDDS contained the aerosol that need to be placed on the NGI or GTI. Indeed, these puffs are combined with a carefully controlled volume of air, and this was used to achieve the targeted flowrate. The vacuum pump helped with mixing the puffs with air to ensure a consistent and uniform blend. For instance, in the NGI, the puffs were introduced into the impactor along with a controlled volume of air, creating a continuous airflow at the specified rate of 15 L/min. Similarly, in the GTI, the mixture of puffs and air was adjusted to attain a flowrate of 60 L/min.

The solutions employed to fill the GTI were also utilized as a recovery method for NGI experiments. In each API experiment, a pre-measured volume (5 mL) of the recovery solution was dispensed into each NGI cup tray using an automatic pipette. This was carried out to dissolve the impacted content on each tray, and subsequent sample analysis was performed with the help of HPLC.

GTI experiments were performed on all devices examined in this study. However, the NGI experiments were exclusively carried out for the nebulizer and VDDS delivering satisfactory respirable doses (Figure 5). Given that pMDIs and DPIs typically have larger MMADs compared to nebulizer, we determined that conducting NGI experiments on pMDIs and DPIs would not be relevant for the purposes of this comparative analysis.

## 5. Conclusions

In conclusion, the differential performances of terbutaline, salbutamol hemisulfate, and ipratropium bromide within the context of drug vaping underscores the critical role played by the unique physicochemical properties of the API in determining their suitability for vaporization.

Terbutaline’s success in the VDDS environment can be attributed to its lower melting point and thermal stability. These attributes facilitated efficient vaporization and minimized the risk of thermal degradation during the process. Consequently, terbutaline proved to be well suited for inhalation through VDDS, having potential as an effective respiratory therapy.

On the other hand, salbutamol hemisulfate and ipratropium bromide presented challenges in the VDDS. The higher melting point of ipratropium bromide posed an obstacle to efficient vaporization, requiring elevated energy input that could potentially lead to thermal degradation. Regarding salbutamol, due to its formulation, the specific VDDS used contributed to its limited compatibility with VDDS.

In summary, it becomes evident that numerous parameters can exert considerable influence on the compatibility of an API with VDDS. These encompass the physiochemical properties of the molecule, including whether it is salt-free; the formulation elements, particularly the composition of the e-liquid employed; and the molecule’s melting point, a critical determinant of its thermal stability. These factors collectively underscore the complexity of optimizing API compatibility for effective drug vaping through VDDS.

## Figures and Tables

**Figure 1 pharmaceuticals-16-01730-f001:**
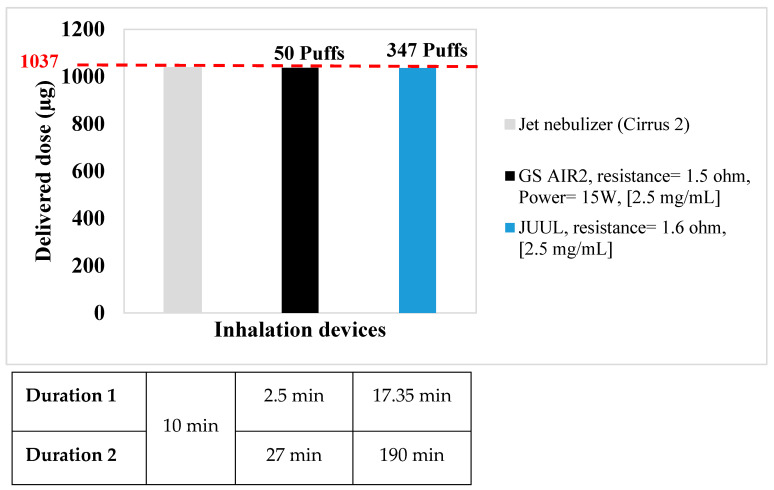
Dose equivalence for terbutaline (comparison between GS AIR 2 and JUUL vs. nebulizer).

**Figure 2 pharmaceuticals-16-01730-f002:**
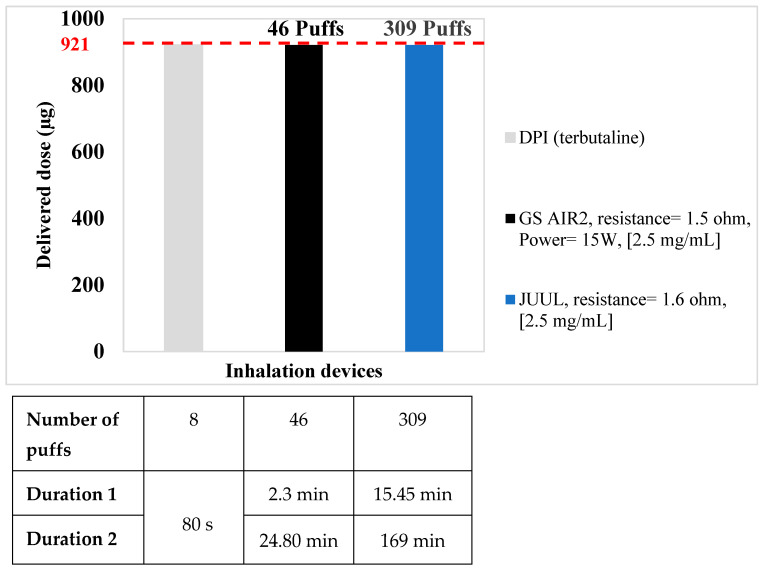
Dose equivalency for terbutaline (comparison between GS AIR 2 and JUUL vs. DPI).

**Figure 3 pharmaceuticals-16-01730-f003:**
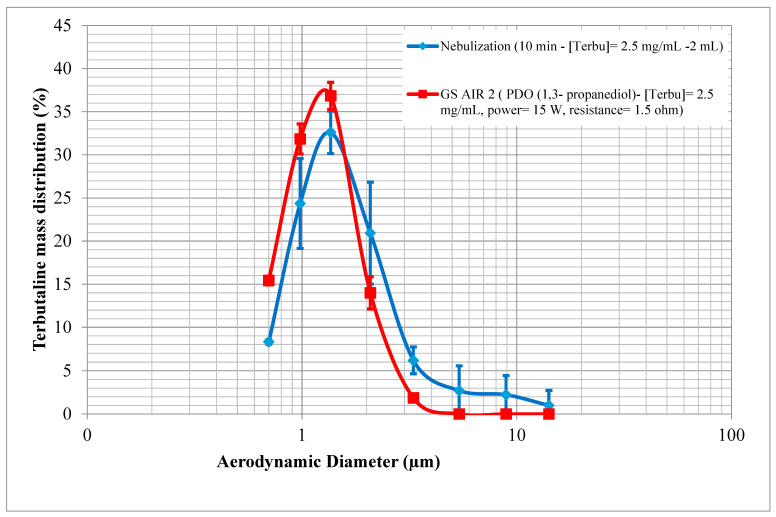
NGI (next-generation impactor) collected data showing the frequency mass distribution for a jet nebulizer and VDDS (GS AIR 2) at a concentration of 2.5 mg/mL, with power set at 15 W and resistance set at 1.5 ohm.

**Figure 4 pharmaceuticals-16-01730-f004:**
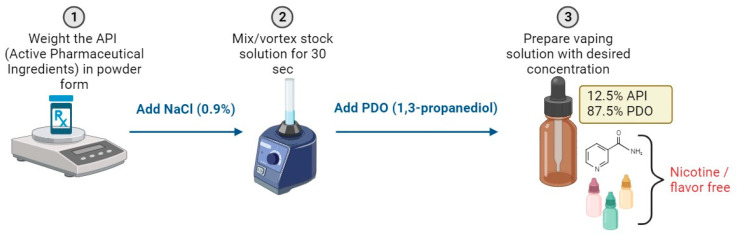
Preparation steps of vaping solution.

**Figure 5 pharmaceuticals-16-01730-f005:**
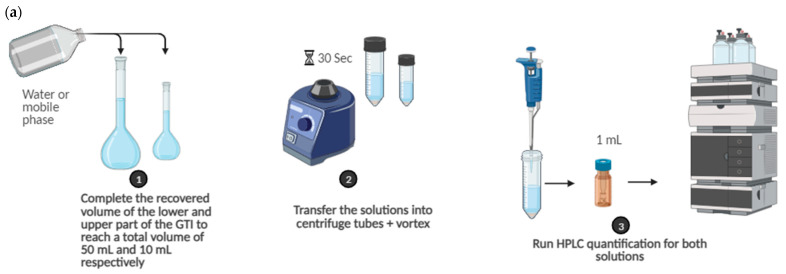
(**a**) Steps followed during GTI (glass twin impinger) experiments. (**b**) Steps followed during NGI (next-generation impactor) experiments.

**Table 2 pharmaceuticals-16-01730-t002:** Optimum nebulization time and volume depending on the tested molecule.

Molecule	Volume (mL)	Nebulization Time (min)
Terbutaline	2	10
Ipratropium bromide	4	20
Salbutamol hemisulfate	2.5	10

## Data Availability

Data are contained within the article and Appendix A.

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
