# Peer review of "Development of a Novel Bronchodilator Vaping Drug Delivery System Based on Thermal Degradation Properties"

_pharmaceuticals, 2023, doi:10.3390/ph16121730_

Round 1
Reviewer 1 Report
Comments and Suggestions for Authors
The article entitled "Bronchodilator Vaping: Investigating Formulation and Drug Selection Based on Thermal Degradation Properties" examined the use of terbutaline, a short-acting β2 agonist (bronchodilator), in combination with VDDS. The study aimed to identify molecules that are suitable for drug vaping, and comprehend the critical factors to consider when selecting optimal drug candidates and aimed to establish a dose equivalence with existing treatments such as nebulizers, pMDI, and DPI. The work is overall good and covers important area. However, there are some shortcomings that prevents its acceptance in the current form:
1- I suggest changing the title to: Development of a Novel Bronchodilator Vaping Drug Delivery System Based on Thermal Degradation Properties
2- The experiment designs and methods needs to be mentioned and clearly presented in full description especially the characterization procedures.
3- The quality of figures 2 and 3 needs to be improved.
4- I suggest including this related important articles in the references to elaborate the discussion of the thermal stability part.
https://doi.org/10.3390/molecules28010306
Comments on the Quality of English LanguageModerate editing
Reviewer 2 Report
Comments and Suggestions for Authors
See the attached file

Reviewer 3 Report
Comments and Suggestions for Authors
The authors have conducted a good piece of study on selecting anti-asthmatic drugs for vaping. However, the outcomes of this study are inconclusive and has a lot of limitations as demonstrated by the authors.
The authors demonstrated that the thermal degradation properties of the active drug are corelated with the vaping technology; however, no data presented supporting this claim. What type of thermal degradation of which drugs studied here? This is a major drawback of the manuscript.
Why is vaping not suitable for all types of drugs need further clarification.
Is vaping feasible for patients with severe asthma or COPD?
Table 2: Delivered doses from GS AIR 2 & JUUL are presented in ug/puff; however, from nebulizers and DPIs or pMDIs, presented in total doses. Are they comparable? How many puffs or inhalation were used to get the delivered doses of terbutaline from nebulizers and DPIs or pMDIs?
Figures 2 & 3 are not required; now a days researchers around the world are familiar with them.
Round 2
Reviewer 1 Report
Comments and Suggestions for Authors
The authors have addressed all comments successfully, so it can be accepted in the present form.
Comments on the Quality of English LanguageModerate editing.
Reviewer 2 Report
Comments and Suggestions for Authors
The authors explained many issues and I accept their responses. The improvements they have made after the review will increase the scientific quality and of this publication.
It is not crucial point, but the authors should notoce that Brownian diffusion is responsible for deposition of submicron and nanosize particles in the lungs, not for their exhalation. This is why inhalation of nanoparticles is potentially harmful. If all particles smaller than 0.5 um (so also nanoparticles) were exhaled, they would not pose a threat.